# Lipid Profiles in Preliminary Germinated Brown Rice Beverages Compared to Non-Germinated Brown and White Rice Beverages

**DOI:** 10.3390/foods11020220

**Published:** 2022-01-14

**Authors:** John C. Beaulieu, Robert A. Moreau, Michael J. Powell, Javier M. Obando-Ulloa

**Affiliations:** 1Food Processing & Sensory Quality Research Unit, United States Department of Agriculture, Agricultural Research Service, Southern Regional Research Center, 1100 Robert E. Lee Blvd., New Orleans, LA 70124, USA; 2Sustainable Biofuels and CoProducts Research Unit, Eastern Regional Research Center, USDA, ARS, 600 East Mermaid, Lane, Wyndmoor, PA 19038, USA; bobmoreauzz@gmail.com (R.A.M.); michael.powell@usda.gov (M.J.P.); 3Doctorate Program in Natural Science for Development (DOCINADE) and Agronomy Engineering School, Costa Rica Institute of Technology (ITCR), San Carlos Technology Local Campus, P.O. Box 223-21001, Ciudad Quesada, San Carlos 30101, Alajuela, Costa Rica; jaobando@itcr.ac.cr

**Keywords:** enzymatic saccharification, functional beverages, germination, lipids, sprouting, value-added

## Abstract

Brown rice is nutritionally superior to white rice, yet oil rancidity can be problematic during processing and storage regarding sensory attributes. Germinating brown rice is known to generally increase some health-promoting compounds. In response to increasing the consumption of plant-based beverages, we sprouted unstabilized brown rice, using green technologies and saccharification enzymes for value-added beverages. ‘Rondo’ paddy rice was dehulled, sorted and germinated, and beverages were produced and compared against non-germinated brown and white brewers rice beverages. The preliminary germinated brown rice beverage contained significantly higher concentrations of total lipids, diacylglycerols, triacylglycerols, free sterols, phytosterol esters and oryzanols than both non-germinated brown and white rice beverages. White rice beverages had significantly higher free fatty acids. Significant lipid losses occurred during sieving, yet novel germinated brown rice beverages contained appreciable levels of valuable health-beneficial lipids, which appeared to form natural emulsions. Further pilot plant investigations should be scaled-up for pasteurization and adjusted through emulsification to ameliorate sieving losses.

## 1. Introduction

Rice feeds approximately half the world’s population and is the main food crop in developing nations [1]. However, the majority of rice consumed is white rice, which is not nutritionally dense and considered a starchy food source. Whole grain brown rice (BR) is superior to white rice (WR) since most nutrients, such as the oils, fatty acids, proteins, vitamins, fiber, micronutrients and antioxidants are retained in the bran [2]. BR containing bran, embryo and aleurone delivers substantial proteins and lipids that convey health-promoting nutritional constituents for consumers [3,4,5,6]. Rice bran also contains high amounts of fiber and bioactive phytochemicals, such as tocopherols, tocotrienols, oryzanols, vitamin B complex, phytosterols (β-sitosterol, campesterol and stigmasterol), carotenoids and phenolic compounds [7]. Several such bioactive compounds have long been recognized to improve human health through antioxidant activities, including scavenging free radicals, enhancing the immune system and reducing the development of cancer and heart disease [8,9,10,11]. Rice flour and rice bran, and certain other grains, are known to contain high levels of lipolytic enzymes that require thermal and non-thermal methods to stabilize these materials [2]. Unfortunately, storage, milling and further food processing affects the lipids, starch and protein, often resulting in undesirable sensory, textural and nutritional changes to the final rice products in the marketplace.

In the past, BR and bran, although nutritious, were usually not consumed because of their high fiber content and possible hull contamination [12], notorious oxidation, off-flavor issues [13] and lengthy cooking time [14]. For marketability and consumer acceptance, stabilization or inactivation of lipase and the inhibition of the formation of free fatty acids (FFA) is considered necessary, immediately after milling. As most older BR protein extraction methods negatively affect the functional and nutritional properties of the proteins, less harsh procedures involving enzymes have been used to extract oil and/or protein from rice flour and bran [2] (pp. 143–162). In full-fat or only partially defatted rice bran, it was noted that the liquid phase could be further processed and stabilized using amylase and amyloglucosidase into rice beverage products [15]. Ironically, the vast majority of plant-based beverages have exogenous oils added toward the end of processing to assist emulsification. Globally, the dairy-alternatives plant beverage market is forecast to surpass USD 34 B by 2024 [16] and, according to the Information Resources Institute, US plant-based beverage sales attained USD 1.7 B, representing a 6.2% increase in one year, through 2019 [17].

Germination (sprouting) is a low-cost technology that initiates with seed water uptake, ultimately followed by the protrusion of the radicle from the seed. Along with strengthened health trends, the advent of “sprouted” whole grain products has markedly increased in the food and beverage marketplace [18]. The content of γ-aminobutyric acid (GABA), free sterols and phytosterol esters, free fatty acids, soluble fiber, γ-oryzanol and antioxidants, such as vitamin E, phenolic compounds and other bioactive compounds, usually increase during BR germination [3,4,5,6,19]. However, there are often different results concerning increased or decreased levels of key nutrients based on the grain type and/or variable soaking and germinating conditions and times [4,19,20,21]. Specifically, the reports addressing the changes in free fatty acids during BR germination also have contradictions [20,21].

Rice bran oil (RBO) offers several health-benefits due to the presence of ferulics, sterols, tocopherols, γ-oryzanol and tocotrienols, which convey antioxidant characteristics and stability along with several health-promoting prebiotic and probiotic benefits [9]. Very few food constituents have been granted the European Commission and the US Food and Drugs Administration approval to use health claims. Plant sterols have attained such approval status due to their proven cholesterol-lowering properties and, recently, attention has refocused on plant sterols regarding the anticarcinogenic and anti-inflammatory effects [7,22]. Subsequently, BR was germinated and assessed for beverage formulations with respect to lipid characteristics. Germinated (sprouted) brown ‘Rondo’ rice (GBR) was softened, wet milled, gelatinized and enzymatically converted into beverages. The methodology has no added oils or salts, additives or fortification. We previously characterized the germination conditions, processing itself, phytic acid and quality parameters in the preliminary beverages [23], along with some key health-beneficial compounds and arsenic levels [24]. GBR beverages should contain significantly higher concentrations of health-beneficial lipids compared to non-germinated brown and white rice beverages. Herein, we investigated this assumption and report several classes of lipids in preliminary beverages from germinated brown rice (GBR), compared to non-germinated brown ‘Rondo’ (BRR) and white rice (WR) beverages.

## 2. Materials and Methods

### 2.1. Rice Material, Germination, Softening, Wet-Milling and Enzyme Processing

Sourcing the ‘Rondo’ rice and methods used to de-hull, mill, sprout, and the novel protocol used to deliver a free-flowing soluble matrix through thermal softening, wet-milling, gelatinization and saccharification, were previously reported [24]. Briefly, ‘Rondo’ was grown at the Dale Bumpers National Rice Research Center (Stuttgart, AR, USA), harvested and dried to 12% moisture, and the paddy rice was cleaned using a screen cleaner (Model MICRO-224-LH, Crippen Northland Superior Supply Co., Winnipeg, MB, Canada) and stored at 60% RH at 4 °C. The same seed lot was dehulled using a Yamamoto Impeller Type Husker (Model FC2K, Calibration Plus, Woodland, CA, USA) and milled into white rice (WR), including broken, Yamamoto Miller Rice Pal (Model VP-32T, Calibration Plus, Woodland, CA, USA) in Stuttgart, AR, then shipped to the Southern Regional Research Center (SRRC) in New Orleans, LA, USA.

Beverage processing methods were optimized for GBR (*n* = 6 per treatment) then compared against non-germinated BRR and WR beverages (*n* = 3 per treatment) using the same methods. For WR, the milled and polished rice was not sorted, which deliberately delivered brewers rice or “seconds” that is often used in commercial beverages. Freshly de-hulled (Satake Husk Aspirator, HA 60B, Higashi-Hiroshima, Japan) sorted and graded (Clipper 400 Office Tester Cleaner; A. T. Ferrell Company, Bluffton, IN, USA) BRR was treated with a peracetic acid food-safety rinse [23], then soaked in a rice:water ratio of 1:1 at 35 °C, followed by germination with rinsing every 4 h, for a total of 48 h, attaining the GBR. Then, WR, BRR and GBR were thermally softened (1:2 rice:water ratio) at temperatures just below the ‘Rondo’ WR gelatinization temperature (<70 °C). After thermal softening, wet-milling in a 4 L blender (Waring Commercial, CB15V, Torrington, CN, USA) with additional water dilution established a free-flowing liquid that additionally avoided gelatinization. Post wet-milling (PWM) samples passed a 30-mesh sieve (0.595 mm or 595 µm, Gilson Co. Inc., Lewis Center, OH, USA). Then, the free flowing beverages were heated to 80 °C for starch gelatinization, followed by liquefaction at ~55 °C using glucoamylase (EC 3.2.1.3) and α-amylase (EC 3.2.1.1) at 300 µL per 100 g starch, followed by sieving through a 140-mesh sieve (0.105 mm or 105 µm, Gilson Co., Inc.). The preliminary beverage (prior to homogenization or pasteurization) was termed as post enzyme (PNZ) beverages. The PWM sieving loss (PWM-SL) and PNZ sieving loss (PNZ-SL) were also analyzed.

Control, pre-processed rice crude fat content were measured [23] and compared, herein, to the total lipids assessed per HPLC (below). Thereafter, the controls were ground into flours using a cyclone sample mill (UDY model 3010-080P, UDY Corporation, Ft. Collins, CO, USA) and all experimental samples were freeze dried into powders and stored at −80 °C for later analyses. No commercial rice beverages were compared since virtually all the products found in local stores had added oils (e.g., safflower and/or canola and/or sunflower) and many also contain additives (fortification). However, two commercial white flours (CRF) were used as comparisons to the in-house developed WR, BRR and GBR flours that were produced after freeze drying, using a cyclone sample mill (UDY model 3010-080P, UDY Corporation, Ft. Collins, CO, USA). A Rivland RL-100 long grain rice flour (Riviana Foods Inc., Houston, TX, USA) and Remyflo R7-150T high amylose rice flour (Remy/Beneo, Morris Plains, Fairfield, NJ, USA) were chosen due to similar characteristics compared to ‘Rondo’ white rice [25].

#### 2.1.1. Lipid Characterization

After all the experiments were completed, stored freeze-dried samples were evaluated for total lipids, free palmitic, free stearic, free linoleic and free linolenic acids, unknown free fatty acids, free sterols, phytosterol esters (which include some very nonpolar lipids), diacylglycerols (DAG), triacylglycerols (TAG) and oryzanol, as described below. Moisture content of each sample was used to calculate the compounds and compound classes on a dry weight basis.

#### 2.1.2. Accelerated Solvent Extraction

The initial samples were present in a ground state as either flours or freeze-dried powders and, therefore, no further grinding was needed. Duplicate 1.0 g samples were prepared. The samples were mixed with uniform 20–30 mesh Ottawa sand (Thermo Fisher Scientific, Waltham, MA, USA) in a beaker and transferred to 11 mL cells. The cells were topped off with Ottawa sand and bottomed off prior to filling; containing cellulose filters at both the top and bottom. Ottawa sand is a very clean, inert, uniformly shaped material that helps to prevent the sample from clumping, and ensures a good flow of solvents in the extraction vessel. The ASE (Accelerated Solvent Extractor, Model 200, Dionex, Sunnyvale, CA, USA) was operated, as previously described [26]. The parameters were as follows: pressure, 1000 psi; temperature, 100 °C; preheat time, 0; heat time; 5 min; static time; 10 min; static cycles, 3; flush volume, 100% (11 mL cell); purge time, 60 sec and the solvent was hexane. All the extracts were dried under a stream of N_2_ in a heated water bath to obtain the total extract weight. For storage, each sample extract was then dissolved in about 10 mg/mL in 85:15 chloroform–methanol with 0.01% butylated hydroxytoluene.

#### 2.1.3. Nonpolar Lipid Analysis

Injections (100 µL) were made of each solvent extracted sample on a Thermo Ultimate 3000 HPLC (Thermo-Fisher, Sunnyvale, CA, USA) using an updated method, as previously reported [27]. The column was a 100 × 3 mm LiChrosorb 5 DIOL (Chrompack, Raritan, NJ, USA), using a 0.5 mL/min flow rate and the following binary gradient: A: 1000:1 hexane:acetic acid; B: 100:1 hexane:isopropyl alcohol with the following ramp: 0 min, 100/0 A/B; 8 min, 100/0; 10 min, 75/25; 40 min, 75/25; 41 min, 100/0 and 60 min, 100/0. Detection was accomplished by diode array detectors (DAD) at 205 nm and 320 nm, and also with the Thermo Charged Aerosol Detector (CAD). Oleic acid, linoleic acid, linolenic acid and the unknown potential free fatty acids (FFA), using linoleic calibration, were appraised at 205 nm and oryzanol at 320 nm. The 254 nm and 280 nm responses were also recorded by the DAD for monitoring, but were not used for any peak analyses. Detection was accomplished by CAD for the steryl esters (StE), triacylglycerols (TAG), stearic/palmitic acid, 1,3-diacylglycerols and 1,2-diacylglycerols (DAG), both using sterol calibration. There were multiple iterations of the various possible DAG compounds, and peaks were therefore broad sets that separated out as 1,2 versus 1,3. The instrument numbers generated were reliable for showing how the classes delivered trends through processing.

### 2.2. Data Analysis and Statistics

Data were initially analyzed to assure a normal distribution using the Shapiro–Wilk W hypothesis test, and outliers were removed to avoid their effect on the results using JMP^®^ 13 PRO for Windows (SAS Institute Inc., Cary, NC, USA). Thereafter, data were analyzed using the analysis of variance (ANOVA) in JMP^®^ 13 PRO for Windows. Comparisons were made across the whole experiment, per treatment, for which there is technically not an overall control (e.g., WR cannot be germinated, there was one non-germinated BRR beverage and another BR was germinated into GBR; Table 1, Table 2 and Table 3). Since the same lot of rice was used for all beverages, data were also presented per rice beverage type (WR, BRR, GBR), based on each unique initial starting material unique control (Appendix A). Throughout, two sets of symbol and letter designations were purposely used in the tabulated data. When statistically significant differences were found, the means were compared against the control by the Dunnett’s test at *p* < 0.05 (illustrated by asterisks). In the cases in which a control was not available, treatment differences were evaluated by the Tukey–Kramer HSD (Honestly Significant Difference) test at *p* < 0.05.

## 3. Results and Discussion

### 3.1. Total Lipid and Proximate Analysis

Total lipids determined by the HPLC from freeze-dried powders indicated that the starting materials contained 0.77%, 2.78%, 3.95% and 2.46% (dry weight basis, dwb), for WR, BRR and BRR sprouted into GBR (often designated as BRR **→** GBR, for clarity) and the GBR, respectively (Table 1). With the exception of the WR controls, these data corroborated well with the original proximate analyses, per trial. The initial crude fat proximate contents, corresponding to the original raw data per trials compared herein, were 1.19 (WR), 3.04 (BRR) and 3.59% in the BRR used for germination, which resulted in 2.48% for GBR [23]. These were the control, pre-processed rice crude fat contents that were potentially delivered into each rice beverage type. However, as discussed below, the original WR proximate analysis was accomplished with the freshly dehulled rice and shipped WR, whereas the later WR lipid determination was accomplished with the samples from stored rice with “brokens” that were afterwards milled into flour.

There was wide variability in the MC among the initial control samples, post-experimental saved samples and commercial flours, due to sourcing, different freeze-drying runs, storage differences and certainly germinated versus non-germinated effects (data not shown). Therefore, a global correction factor was applied to all the data by calculating each result based on the unique MC of the samples, resulting in the delivery of data as percentage lipid constituents (mg/100 g, dwb). The WR PWM-SL (sieving loss) was negligible (1.05%) and there was an insufficient sample quantity to collect and analyze (Table 1). The GBR PNZ-SL (sieving loss) samples were unfortunately not collected or lost. Lipids were not measured in any commercial rice beverages because every product found on the grocer’s shelf had exogenously added oil (labeled as canola and/or safflower and/or sunflower oil), which would confound data regarding the endogenously present oils and impart obvious lipid, FFA and sterol impurities [22].

### 3.2. Total HPLC Lipids (and Oils) versus Summed Recovered Compounds

When one compares the total lipids/oil (% dwb) to the summed amount of the compounds recovered (mg/100 g, dwb), the initial HPLC percentages were very congruent with the overall summed lipids recovered and reported (Table 1 and Table 3). Due to the utilized units, there is a 1000-fold numerical difference between % dwb total (HPLC lipids) versus the summation of all the recovered compounds and compound classes (mg/100 g). In the 3 PNZ beverages, there was an overall 93.2 ± 9.0% lipid/oil recovery rate. Aside from some FFAs, the levels of total lipids in the GBR PNZ beverages were oftentimes significantly higher than the non-germinated BRR beverage, which was likewise significantly higher than the WR beverage (Table 1, Table 2 and Table 3). The WR beverage stream contained the least amount of starting lipids and also lost the most significant proportion through processing into the WR PNZ beverage, compared to both the BRR and GBR beverages (Table 1, Table 2 and Table 3; Appendix A).

An Italian plant-based beverage survey evaluating 72 commercial rice beverages found the average oil content of 1.0 g/100 mL, with a range from 1.0–1.1 g/100 mL with saturates comprising 0.2 g/100 mL [28]. This level corroborates with the level reported in unsweetened rice beverages in the USDA Food Data Central database of 0.97 g (https://ndb.nal.usda.gov/fdc-app.html#/food-details/171942/nutrients) (accessed on 14 September 2021). However, this was a back-calculated value and it is unknown if oil was added in this sample. Oil levels reported elsewhere in commercial samples (generally about 1 g/100 mL) are not on par with the levels found herein (where no exogenous oils were added) in post-enzyme treated (PNZ) WR and BRR beverages, but were closer to the completely endogenous GBR PNZ beverage, containing 0.62 g/100 g (Table 1). Most plant-based beverages are processed with stabilized raw ingredients, which effectively strip away endogenous and natural lipids that cause rancidity issues. Herein, we purposely used native, non-stabilized brown rice that was sprouted and conveyed most endogenous ingredients back into a processed beverage using “green technologies”. Subsequently, the 0.62% oils contained in the GBR PNZ beverage is 100% natural, and requires no added ingredients or exogenously added oils to augment emulsion and, hence, has no labeling considerations.

The overall percentage of the material lost during the process through sieving, followed the trend whereby BRR > GBR > WR [23] and PNZ GBR samples that passed through a 140-mesh sieve had a significantly larger mean and D90 cumulative particle size, compared to both the BRR and WR PNZ beverages [24].The GBR PNZ had about half the processing loss compared to the non-germinated BRR starting material, while the WR process resulted in the least losses and better solubilization as almost the entire starting weight was starchy material [23]. Nonetheless, where lipid processing losses were reported (PWM-SL and PNZ-SL), these values were often the most significant percentage of the materials recovered, per beverage type, on a dry weight basis (Appendix A). For example, there was a 5.17% loss of total lipids in the GBR PWM. It is of note that these sieving loss steps inherently have a “concentrating effect”, as very low MC residuals (e.g., starchy granules and fiber) have been removed by the sieves during the processing regime, whereas a 5-fold dilution has carried forward the liquid matrix into the resulting beverages.

### 3.3. Free Fatty Acids (FFA): Saturated (SFA) and Unsaturated (USFA)

The major free fatty acids recovered in the three starting materials (WR, BRR and BRR that was germinated into GBR) were palmitic/stearic, oleic and linoleic acids (Table 1), as generally reported in most rice and BR fractions [2] (pp. 163–190). The method used herein did not fully resolve the two saturated fatty acids, stearic versus palmitic. However, stearic acid generally comprises only 2–4% of the total lipid profile in rice [2] (pp. 163–190). Total FFAs recovered in WR, BRR and GBR were 523.06, 53.15 and 47.80 mg/100 g, respectively (Table 1). In general, there was a fairly even distribution of the 3 main FFA categories across controls and through processing, whereby roughly 20 to 40% was comprised by palmitic (C16:0)/stearic (C18:0), oleic (C18:1) and linoleic (C18:2) acids. Similar ranges were reported in rice bran oil (RBO) [29]. The palmitic/stearic, oleic and linoleic acids composition based on dry weight for the major of the fatty acids recovered in BRR, was 21.59, 15.74 and 13.18 mg/100 g, and in the sprouted method the BRR was 29.98, 27.67 and 26.64 mg/100 g, which gave rise to the GBR containing 24.69, 11.83 and 9.44 mg/100 g, respectively (Table 1).

Germination caused significant decreases in all the free fatty acids evaluated (Table 1). During rice germination, the oleic acid decreased, whereas the palmitic and linoleic acid contents increased [20]. On the other hand, the concentrations of oleic, palmitic and palmitoleic acids increased in the initial stage of germination, but decreased rapidly after 72 h [21]. However, both of these studies measured the fatty acids after transesterification, so they were measuring the esterified fatty acids (in glycerolipids, such as triacylglycerols, glycolipids and phospholipids), whereas the individual free fatty acids were evaluated herein. Furthermore, it appears that the methodology used by [21] to measure FFA would combine the data with other lipids, which also makes comparisons to our data difficult. In another study, the contents of the oil components (palmitic, oleic and linoleic acid), γ-oryzanol, phytosterol, vitamin E and squalene were slightly increased or not changed by germination in the two rice varieties [30]. Nonetheless, in the germination portion of the study, virtually every FFA significantly decreased after the 48 h germination period, and further decreases were generally observed as beverage processing (control → PWM → PNZ) ensued (Table 1).

In general, the ‘Rondo’ WR controls had significantly higher concentrations of all the FFAs in this study (aside from one reference commercial rice flour, CRF) and the significantly lower total lipid recovered compared to the BRR and GBR controls (Table 1). The WR contained 138.98, 199.17 and 178.59 mg/100 g, palmitic/stearic, oleic and linoleic acids, respectively (Table 1). The two commercial white flours that were included for comparison, likewise, had very high levels of FFAs. Nonetheless, it is not legitimate to make statistical comparisons between unknown commercially processed flour versus a well-characterized experimental variety and process. FFA levels in both CRF’s were remarkably higher (232.7–645.8 mg/100 g) than the freshly de-hulled ‘Rondo’ non-germinated BRR samples (11.3–53.2) and GBR samples, (29.9–87.9), discounting the loss streams (Table 1). Subsequently, the lipids in the endosperm of the WR experienced a fair amount of lipase activity. Although 523 mg FFA/100 g appears to be a high amount, it only translates to 0.523%. Therefore, in the WR, this quantity of FFAs is not really a large proportion of the overall nutrients. As the total lipid in white rice was 774 mg/100 g, then the FFAs appeared to be the most abundant lipid in these WR samples (Table 1).

Prior to converting all the data to a dry weight concentrations basis using the discrete sample MC, the control BRR and GBR samples used herein had free fatty acids (FFAs), free sterols and diacylglycerols (DAGs) levels (data not shown) within the similar ranges previously reported in the control ground “Macia” sorghum [31]. These levels of FFAs indicated higher than normal levels of lipase activity in the sorghum, as was likewise probable in the WR findings. The materials used to produce the WR beverages and stored commercial white flours (CRF) used as a comparison had excessive FFAs. It has long been known, however, that rice (even though it contains a relatively low amount of oil) and RBO, are subject to the rapid accumulation of FFA and lipid oxidation products, due to the exceedingly high lipase levels, even in the mature, dried kernels [2,32] (pp. 143–162).

These data indicate that the WR experienced substantial lipid hydrolysis from the time of milling through shipping/handling and short 4 °C storage (1 month) before use, or due to the sample freeze drying and flour produced thereafter. However, brokens were purposely received and used to make a low cost, value-added beverage. On the other hand, all the BRR and GBR samples were freshly dehulled and immediately utilized, unstabilized, for each experiment, then frozen prior to freeze-drying and flour/powder sample production. Rice has one of the most notoriously active and persistent levels of lipase activity and this apparently resulted in the initial free fatty acid differences during these trials. However, these experiments were optimized for the GBR, and the non-germinated BRR and WR were run as “checks” and helped serve as comparison and for validation purposes.

The individual FFAs, across all three beverages, decreased from controls (WR or BRR or GBR) after wet-milling (PMW), but generally increased after saccharification (PNZ) (Table 1). There was a substantial and significant loss of most FFAs in the WR samples, as controls were heated (softened), wet milled into PWM and enzyme-treated resulting in the PNZ beverage (Table 1). Significant overall losses also occurred oftentimes in both the BRR and GBR beverages. When the values were converted from mg/100 g to relative percentage, the total FFA loss in WR PNZ was 85.5%, without concomitant increase measures in any other analyte to compensate for the mass balance. Total FFA losses in BRR were less (70.7%) and the total FFAs in GBR increased (15.8%), as linoleic acid interestingly increased through processing from PMW to PNZ in all three beverages. Perhaps this is a result of the second heating step (80 °C), in which the starch is purposely hydrated through gelatinization to physically facilitate the saccharification enzyme process, which dissociated TAG and DAG to free more fatty acid moieties. In RBO from the germinated rice, linoleic and linolenic acid composition increased while oleic and palmitic acid decreased [33], which mirrors our trends observed from the GBR controls into the PNZ beverage. Regardless, this is a positive finding since linoleic acid is an essential FA. Overall, there was a general trend in all three beverage types (WR, BRR and GBR), whereby there were marked decreases in almost all the compounds (such as the aforementioned TAG and DAG), except FFAs, through processing (control → PWM → PNZ).

### 3.4. Triacylglycerols (TAG) and Diacylglycerols (DAG)

The BRR used for germination, GBR (BRR **→** GBR) and BRR controls contained the significantly highest level of TAGs in all the analyzed samples, with 3163.5, 1919.2 and 2356.2 mg/100 g, respectively, and 140.7, 62.2 and 103.6 mg/100 g DAGs (1,3-DAG plus 1,2-DAG), respectively (Table 2). All the samples utilizing BRR and GBR from control through the PNZ beverages (not including the processing losses PMW-SL and PNZ-SL) contained between 81.6–92.6% of the recovered lipids as TAGs and DAGs (Table 4). Before converting data to dwb, approximately 70–80% (wet wt%) of the compounds recovered in the BRR and GBR samples were TAG and DAG, similar to the levels previously reported in RBO [32]. On the other hand, the WR beverage control contained significantly less TAG, 1,3-DAG and 1,2-DAG compounds (167.89, 13.39 and 3.40 mg/100 g, respectively) and markedly reduced the relative percentages (24.6–72.9%) throughout processing (Table 2 and Table 4). This indicates that relatively low lipase or oxidation occurred in the BRR and GBR samples compared to the WR treatments. Lipase hydrolysis of esterified fatty acids (FA) from oil triacylglycerols (TAG) produce 1,2,diacylglycerols (DAG) and 1,3,diacylglycerols (DAG), which ultimately leads to the net conversion of oil to sugars during germination [34]. The breakdown in TAGs was the main expected change in FFAs to occur during germination. Indeed, from BRR **→** GBR, there was a significant decrease in TAGs, 1,3-DAG and 1,2-DAG of 39.3, 58.5 and 55.2%, respectively (Table 2). Except for 1,3-DAG in GBR PWM, the TAGs and DAGs significantly decreased from controls through PWM (presumably due to heating and wet-milling) into each rice type PNZ beverage (Appendix A).

The WR samples and CRF, in general, displayed the same trends regarding the classes of FFAs and TAGs/DAGs having high or low concentrations. WR control samples (aside from CRF) had the lowest significant concentration of TAGs and DAGs in the study (Table 2). This appears to be consistent with the FFAs being starch lipids. However, when total TAGs and DAGs were expressed as a relative percentage of the total lipid compounds recovered, the control WR, BRR and GBR contained 24.6, 92.0 and 91.0%, respectively (Table 4). This was due to the fact that the WR had the highest significant quantity (523.06 mg/100 g; Table 1) and relative percentage (69.6%; Table 4) of FFAs, compared to the BRR and GBR. The majority of the total lipids recovered in BRR and GBR were TAGs (92.0 and 91.0%, respectively; Table 4), which concomitantly contained relatively low levels of total FFAs (2.0 and 2.2%, respectively). These results paralleled the above FFA finding, indicating that the WR (as well as stored WR check flours) succumbed to lipid oxidation. Aside from a few exceptions (linoleic acid, linolenic acid and 1,3-DAG in GBR), there were generally significant decreases in most of the parameters measured (FFAs, TAGs, 1,2-DAG, sterols and oryzanols) in all three beverage processes, especially in non-germinated BRR and WR (Appendix A).

### 3.5. Phytosterol Esters (StE) and Nonpolar Lipids

Similar to the other compound trends, the GBR contained significantly lower levels of phytosterol esters (97.29 mg/100 g), compared to the original BRR starting material (138.50 mg/100 g). Again, there was a general trend whereby the processing caused an initial decrease (in PWM samples) and/or significant decrease in the phytosterol esters in each PNZ beverage (Table 3). In a study looking at the compositional change of policosanols and oils in four varieties of post-germinated brown rice oil, squalene increased 2.4 fold and the phytosterols campesterol, stigmasterol and β-sitosterol increased by 8.3%, 31.6% and 3.3%, respectively, whereas the cycloartenol and 24-methylcycloartanol (probably from the hydrolysis of γ-oryzanol) decreased by 11.0 and 4.5%, respectively [33]. Herein, the phytosterol esters were measured but the peak actually contains other very nonpolar lipids, such as hydrocarbons (including squalene) and wax esters. For example, we reported between 20.4–138.5 mg/100 g of phytosterol esters in the BRR and GBR processing stages, whereas others [22] reported 4.3 mg/100 mL for the rice beverage (principally β-sitosterol, β-sitosterol-β-D-glucoside, campesterol and stigmasterol). The unstabilized 100% natural PNZ beverages delivered 19.3, 20.4 and 29.9 mg/100 g in WR, BRR and GBR, respectively, with the GBR PNZ beverages being significantly higher than the non-germinated beverages (Table 3). Technically, these phytosterol esters can be better classified as “very nonpolar lipids”. Although the analytical system herein utilized did not separate the phytosterol esters from the other non-polar lipids, we believe there is still value demonstrating this peak since interesting trends were observed in these health-promoting compounds [7,22]. The beverage processing loss streams (PWM-SL and PNZ-SL) contained the highest significant levels of phytosterol esters per beverage category tested (Table 3).

### 3.6. Free Sterols

Brown ‘Rondo’ rice (BRR) contained 35.30 mg/100 g total free sterols, which significantly decreased upon germination to 26.92 mg/100 g in GBR. The other BRR control that was not germinated contained 22.28 mg/100 g and the significantly lowest free sterol level was found in the WR control (3.35 mg/100 g) (Table 3). Through processing, the total free sterols significantly decreased to 8.74, and 3.43 and 2.03 mg/100 g in the post-enzyme treated (PNZ) GBR, BRR and WR, respectively. The concentration in the non-germinated WR and BRR control (PNZ) beverages were on par, compared to the 4.29 mg/100 g reported [22]. However, the exact constituents of those beverages tested (e.g., BR versus sprouted, organic, or what commercial processes were employed; [22]) was not determined. The germinated BRR, used in the free-flowing green process to generate GBR PNZ beverages, contained significantly greater concentrations of total free sterols (8.74 mg/100 g) than the non-germinated beverages and the aforementioned well-characterize rice beverage, with 4.29 mg/100 g [22]. This is a valuable finding regarding the health-related advantages of utilizing this methodology to deliver a fully 100% natural plant-based beverage. Due to the analogous structure of cholesterol, many phytosterols are known to compete and interfere with the absorption and binding of cholesterol in the GI tract, ultimately decreasing low-density lipoprotein (LDL) cholesterol levels, which can decrease the threat of coronary heart failure [7,22]. GBR PNZ delivered the significantly highest level for all the compounds listed in Table 2 and Table 3 (TAG, DAG, sterols, oryzanol and the summation of all lipids reported), whereas both the non-germinated BRR and WR beverage quantities were markedly lower. These classes of compounds in the non-germinated BRR PNZ were 1.4- to 5.1-fold lower than the GBR PNZ beverage. Furthermore, the WR PNZ beverage values were even lower, at 1.5- to 16.2-fold lower than GBR PNZ. As with the other assessed lipid categories, the free sterol loss stream (PWM-SL and PNZ-SL) contained the highest significant levels recovered (on a dry weight basis) in each beverage type (Appendix A).

### 3.7. Oryzanol

Significantly lower oryzanol levels were found in WR (5.66 mg/100 g), compared to BRR (35.67 mg/100 g) and the initial BRR (51.62 mg/100 g) used to sprout the GBR (25.10 mg/100 g) (Table 3). Oryzanol levels followed a trend whereby control BRR > GBR > WR, with 0.8, 1.3, 1.5 and 1.2% of all lipid recovery attributed to oryzanol in WR, BRR and BRR → GBR, respectively (Table 4). Oryzanol levels in BRR and GBR compared well to those in 30 BR varieties grown at different sites and in different seasons that delivered an average 26–63 mg/100 g of γ-oryzanol [35], in 16 Korean rice varieties displaying a range of 26.7–61.6 mg/100 g [36], and in a summary of 59 whole grain BR varieties [37]. The oryzanol concentrations in BRR were 6.3- to 14.5-fold and GBR was 7.6-fold higher than the ‘Rondo’ WR. Other BR and GBR have been found to contain roughly 5 times more γ-oryzanol than the counterpart polished rice in “Heugkwang” (black rice) and “Keunnun” (giant embryo) [30]. Significantly lower oryzanol concentrations were recovered in WR, and the CRF samples had the lowest oryzanol levels recovered (1.22 and 3.28 mg/100 g) compared to the GBR and BRR (Table 3).

Sprouting conditions used in ‘Rondo’, resulted in a 2-fold reduction in the oryzanols (Table 3). However, the concentration of γ-oryzanol was previously reported to increase [5], or did not markedly change through the various germination protocols [19,30]. Total oryzanol concentrations conveyed forward into each PNZ beverage also significantly decreased to 5.84, 1.14 and 0.36 mg/100 g in GBR, BRR and WR, respectively. Oryzanol is an ester and it is therefore possible that it can be hydrolyzed by lipases, or degraded as a consequence of the two heating steps during the beverage formation. Researchers have already indicated that the methods to maximize the concentration of γ-oryzanol in germinated rice needs further investigation because initial levels are variety-dependent [35] and germination conditions (the duration and rates of water uptake) are known to affect the metabolic mobilization and concentration changes in this class of compounds during sprouting [5]. In the future, a time-course evaluation of γ-oryzanol and other key phytonutrients is warranted to better optimize the germination stopping point.

All processing loss stages recovered and analyzed had significantly higher levels of oryzanol (Table 3 and Appendix A). The PWM GBR loss was 165.86 mg/100 g. Unfortunately, the GBR PNZ loss samples were not collected or analyzed to tally the overall oryzanol loss. However, the 2 sieving stages in BRR resulted in 201.5 mg/100 g of lost oryzanol, whereas only 11.42 mg/100 g of oryzanol was lost in the WR PNZ (there was negligible WR PWM-SL and, thus, no samples). Particle size of the materials lost and discarded from the sieves was not measured. Yet, the WR losses (principally gritty, starchy endosperm) had the lowest relative percentage loss, whereas the losses in the BRR were the highest while GBR losses were intermediate. This indicates that the germination and endogenous enzymatic activity must have softened, and solubilized more constituents in the GBR beverage stream compared to the non-germinated BRR beverage [23]. As previously noted, the BRR and GBR materials were slightly gritty and brown, indicating that some fiber-associated bran and aleurone materials and hard starchy endosperm constituents were discarded. This implies that, across the board, the processing regime either failed to sufficiently soften/wet-mill the raw materials and solubilize the majority of oryzanols into the beverages, or sieving needs to be readdressed. For example, 25.5, 3.2 and 6.3% of the original control oryzanol was conveyed into the GBR vs. BRR vs. WR beverages, respectively (calculated from Table 1). Only the germinated, endogenously softened, enzyme-activated GBR materials conveyed a significantly higher relative percentage of the oryzanols into the beverages. This trend was observed throughout all three beverages regarding most recovered lipid categories and compounds.

γ-Oryzanol is technically a mixture of ferulic acid esters of triterpene alcohols and sterols. More specifically, these hydroxycinnamate sterol esters are esters of cycloartenol and 24-methylenecycloartanol in rice and sitostanol in corn, which have also recently been demonstrated to contain coumaric, caffeic and sinapic acids esterified to sterols in rice and corn [7]. The ferulate part of γ-oryzanol is attributed to the antioxidant capacity in this sterol class, according to a linoleic acid model wherein the major compounds of γ-oryzanol (cycloartenyl ferulate, 24-methylenecycloartanyl ferulate and campestanyl ferulate) prevented the ultraviolet-derived oxidation of linoleic acid, although the effects were less pronounced than free ferulic acid and α-tocopherol [9], and make up about 90% of the γ-oryzanol in GBR [30]. Hence, one can make an assertion that several additional health-beneficial low molecular weight compounds are probably made available and remain soluble in the GBR beverage prepared and reported herein. Subsequently, more complete softening, particle size reduction and/or emulsification should be employed in the general processing scheme to ameliorate these losses.

Rice bran contains ferulic acid in an insoluble bound form that is esterified with arabinose or arabinoxylans as feruloylated arabinoxylo-oligosaccharies [38]. Excellent functional and emulsifying oil-in-water properties have been attributed to arabinoxylans in grains [39] due to the cross-linking of their ferulic acids [40], which have a unique capacity to form covalent gels [41]. In this beverage production system, decreasing the starch content through native and exogenous saccharification enzymes, generating lower molecular weight oligosaccharides, would also aid to stabilize an emulsion. We previously speculated how the sugars (~15%), oligosaccharides, fiber, protein and oils remaining in these GBR beverages can lead to a natural emulsion [24]. We believe that the neutral pH, along with endogenously generated catabolically produced polysaccharides and exogenously delivered enzymatic oligosaccharides, interacted positively with the proteins to inhibit oil aggregation, which can lead to undesirable feathering and sedimentation. In short-term GBR PNZ storage, no sedimentation was observed [23,24]. Study of the rheological profiles and particle size will continue in scaled-up pilot plant studies, including emulsification followed by pasteurization.

## 4. Conclusions

A tenet of the research conducted on the rice beverages presented herein was to use “green technologies” and rice materials that were not stabilized by any chemical or physical treatments, prior to using a “free-flowing” natural (aside from food grade saccharification enzymes) value-added process, using no additives, oils or salt. Older beverage patents and technologies have included steps for the stabilization and/or rice protein, or oil extraction methods relying heavily upon chemical (acid or base) processes that oftentimes negatively affect the functional and nutritional properties of the proteins and remove endogenous oils. Enzymatic methods are also available and likely well suited, yet an enzyme cocktail is needed if the bran/germ has not been removed or germinated. Much of the data presented herein illustrates the significant lipid losses through germination and processing, especially attributed to sieving, which would be desirable to keep in the pipeline through product development. Nonetheless, this report documents how preliminary GBR PNZ beverages contained significantly greater concentrations of total lipids, TAGs, DAGs, free sterols, phytosterol esters and oryzanols, than both non-germinated BRR and WR PNZ beverages. These are valuable findings, considering the possible health-promoting compounds identified and discussed. Free sterols, phytosterols and oryzanol recovered at significantly higher concentrations in the GBR beverage are promising, concerning recently advancing knowledge regarding both the compound characterization and relevance to human health and well-being.

The developed GBR beverage method has low inputs, requires relatively simple and inexpensive equipment and is applicable for both germinated brown and colored rice varieties. Based on the observations and physicochemical data, a significant amount of valuable endogenous lipids are retained in the GBR beverage, which appear to be incorporated into a natural emulsion. In this beverage process, the heat plus oligosaccharides can provide good conditions to form emulsions with the FFA, lipids and bran-associated fiber, protein and reactive antioxidant compounds. Both sieving steps on a 30-mesh sieve (PWM-SL) or 140-mesh sieve (PNZ-SL) resulted in major fiber-related, lipid and protein losses. Subsequently, the process itself needs to be refined to better soften and wet-mill the starting materials. Future work should analyze the arabinoxylans, ferulic acid containing compounds and ferulates, soluble fiber, and characterize better the oryzanols and phytosterols through emulsification and pasteurization in these newly developed all-natural beverages. Modified methods that capture all the previously documented sieving losses and pilot plant scale-up, should position this advantageous green processing methodology to deliver 100% natural, no additives, value-added germinated rice beverages. This is important industrially and economically, considering the burgeoning plant-based beverage market and the desire of industries to capture more non-animal protein and health-related attributes from an agronomic and relatively inexpensive crop, such as rice. Developing plant-based, protein- and lipid-rich functional beverages with rice that has proven health benefits, will have a positive economic impact.

## Figures and Tables

**Table 1 foods-11-00220-t001:** Total lipids (weight %, dwb) and free fatty acids (mg/100 g) in three beverages prepared from white, brown and germinated brown ‘Rondo’ rice, with commercial flour comparisons.

Treatments	Total Lipid/Oil Weight % (dwb)	Palmitic and Stearic	Oleic	Linoleic	Linolenic
*BRR (→ GBR)* ^1^	*3.95 ± 0.72 z* ^2^	*29.98 ± 5.64 z*	*27.67 ± 10.45 z*	*26.64 ± 4.46 z*	*1.61 ± 0.56 z*
GBR (control)	2.46 ± 0.18 yb	24.69 ± 1.58 yb	11.83 ± 5.69 yb	9.44 ± 1.49 yb	0.50 ± 0.14 yb
BRR (control)	2.78 ± 0.15 a	21.59 ± 3.62 b	15.74 ± 5.59 b	13.18 ± 1.88 b	0.71 ± 0.20 b
WR (control)	0.77 ± 0.09 c	138.98 ± 29.41 a	199.17 ± 36.27 a	178.59 ± 35.77 a	6.09 ± 1.42 a
GBR, PWM	1.98 ± 0.04 *B	14.45 ± 1.72 *A	5.37 ± 1.31 *A	8.81 ± 1.30 A	0.52 ± 0.05 A
BRR, PWM	2.17 ± 0.16 *A	6.85 ± 1.07 *B	1.18 ± 0.57 *B	1.51 ± 0.34 *C	0.16 ± 0.09 *B
WR, PWM	0.14 ± 0.02 *C	5.93 ± 0.45 *B	7.09 ± 0.37 *A	5.57 ± 0.79 *B	0.18 ± 0.05 *B
GBR, PNZ	0.62 ± 0.12 *t	20.85 ± 2.02 *s	6.72 ± 1.56 *s	26.03 ± 3.69 *t	1.40 ± 0.20 *t
BRR, PNZ	0.23 ± 0.02 *s	5.61 ± 1.01 *r	1.91 ± 0.42 *r	7.40 ± 1.03 *s	0.43 ± 0.08 s
WR, PNZ	0.14 ± 0.00 *s	31.46 ± 0.51 *t	13.87 ± 4.06 *t	29.07 ± 4.56 *t	1.33 ± 0.24 *t
GBR, PWM-SL	4.77 ± 0.41 *Z	40.32 ± 3.36 *Z	16.14 ± 2.02 *Z	33.66 ± 3.02 *Z	1.98 ± 0.10 *Z
BRR, PWM-SL	3.93 ± 0.16 *Y	18.96 ± 1.39 Y	3.92 ± 0.72 *Y	5.90 ± 0.21 *Y	0.42 ± 0.01 Y
WR, PWM-SL	i.s.^3^	i.s.	i.s.	i.s.	i.s.
GBR, PNZ-SL	n.s.^3^	n.s.	n.s.	n.s.	n.s.
BRR, PNZ-SL	3.26 ± 0.09 *T	66.05 ± 3.01 *S	24.05 ± 3.04 * S	116.84 ± 1.71 *S	5.83 ± 0.09 *S
WR, PNZ-SL	0.94 ± 0.10 S	123.49 ± 28.11 T	82.50 ± 19.97 * T	170.52 ± 23.43 T	7.35 ± 0.61 T
CRF R7-150T	0.33 ± 0.01	85.13 ± 7.78	72.91 ± 6.52	71.72 ± 7.83	2.05 ± 0.19
CRF RL-100	0.89 ± 0.01	189.2 ± 19.81	226.08 ± 24.10	223.24 ± 24.58	6.58 ± 0.32

^1^ Treatment acronyms: BRR, brown ‘Rondo’ rice; GBR, germinated brown rice; PWM, post wet-milling; PNZ, post enzymes; PWM-SL, post wet-milling sieving loss; PNZ-SL, post enzymes sieving loss; WR; white rice and CRF, commercial rice flour. The *BRR (→ GBR)* is italicized since it was the original starting material to generate GBR but, it is technically not the GBR beverage control. ^2^ Means highlighted with an asterisk (*) are significantly different from the rice type control (GBR, BRR or WR) according to a Dunnett’s test at *p* < 0.05. Control and treatment means not connected by the same letter are significantly different among them, according to a Tukey–Kramer HSD test at *p* < 0.05. z, y indicates the significant differences between the germinated GBR control versus the initial BRR used for germination; a, b, c indicates the differences among the GBR, BRR and WR controls; A,B,C indicates the differences among GBR, BRR and WR for the PWM treatments; X,Y,Z indicates the differences among GBR, BRR and WR for the PWM-SL treatments; r, s, t indicates the differences among GBR, BRR and WR for the PNZ treatments; and R,S,T indicates the differences among GBR, BRR and WR for the PNZ-SL treatment. The data represent the means from independent comparisons, where *n* = 3 or *n* = 6 ± standard deviation. ^3^ i.s. indicates insufficient sample to collect, whereas n.s. indicates not sampled.

**Table 2 foods-11-00220-t002:** Free fatty acids and acylglycerols (mg/100 g) in the three beverages prepared from white, brown and germinated brown ‘Rondo’ rice, grouped by treatment, with commercial flour comparisons.

Treatments	Unknown FFA	Total FFAs	TAG (Triacylglycerols)	1,3-DAG (Diacylglycerols)	1,2-DAG (Diacylglycerols)
*BRR (→ GBR)* ^1^	2.04 ± 1.00 z ^2^	87.93 ± 19.59 z	3163.52 ± 31.50 z	24.04 ± 7.98 z	116.62 ± 24.92 z
GBR (control)	1.33 ± 0.46 yb	47.80 ± 6.86 yb	1919.15 ± 6.86 yb	9.97 ± 4.78 yb	52.24 ± 9.66 yb
BRR (control)	1.94 ± 0.24 a	53.15 ± 10.86 b	2356.24 ± 205.96 a	23.49 ± 6.48 a	80.16 ± 15.93 a
WR (control)	0.23 ± 0.04 c	523.06 ± 102.37 a	167.89 ± 27.23 c	13.39 ± 4.15 b	3.40 ± 0.48 c
GBR, PWM	0.70 ± 0.12 *B	29.85 ± 3.02 *A	1583.44 ± 67.07 *B	15.05 ± 3.00 *B	64.48 ± 10.33 A
BRR, PWM	1.64 ± 0.17 A	11.33 ± 2.00 *B	1844.73 ± 104.49 *A	32.03 ± 3.06 A	43.49 ± 6.32 *B
WR, PWM	0.08 ± 0.01 *C	18.85 ± 0.96 *B	94.31 ± 15.93 C	8.73 ± 2.74 C	3.13 ± 0.81 C
GBR, PNZ	0.33 ± 0.06 *t	55.33 ± 5.51 s	438.85 ± 82.06 *t	9.37 ± 1.33 t	7.84 ± 1.60 *t
BRR, PNZ	0.20 ± 0.02 *s	15.55 ± 2.44 *r	171.39 ± 26.26 *s	5.39 ± 1.19 *s	2.24 ± 0.55 *s
WR, PNZ	0.22 ± 0.02 t,s	75.95 ± 8.37 *t	39.59 ± 12.53 *s	3.64 ± 1.59 s	1.71 ± 1.07 s
GBR, PWM-SL	3.66 ± 0.41 *Y	101.53 ± 4.84 *Z	3697.75 ± 403.22 *Z	19.61 ± 3.58 *Y	177.72 ± 11.82 *Z
BRR, PWM-SL	4.64 ± 0.28 *Z	33.841.15 *Y	2999.40 ± 4.47 *Y	36.56 ± 5.95 *Z	99.90 ± 16.28 Y
WR, PWM-SL	i.s. ^3^	i.s.	i.s.	i.s.	i.s.
GBR, PNZ-SL	n.s. ^3^	n.s.	n.s.	n.s.	n.s.
BRR, PNZ-SL	2.37 ± 0.18 *T	215.15 ± 3.25 *S	2578.79 ± 241.27 T	107.36 ± 13.16 *T	36.42 ± 8.67 *T
WR, PNZ-SL	0.82 ± 0.08 *S	384.68 ± 70.68 T	410.12 ± 83.03 *S	46.63 ± 11.04 *S	16.21 ± 2.43 *S
CRF R7-150T	0.89 ± 0.03	232.71 ± 22.06	18.32 ± 2.28	0.51 ± 0.13	0.56 ± 0.06
CRF RL-100	0.68 ± 0.14	645.80 ± 68.64	159.14 ± 16.45	6.38 ± 2.54	3.38 ± 0.43

^1^ Treatment acronyms: BRR, brown ‘Rondo’ rice; GBR, germinated brown rice; PWM, post wet-milling; PNZ, post enzymes; PWM-SL, post wet-milling sieving loss; PNZ-SL, post enzymes sieving loss; WR; white rice and CRF, commercial rice flour. Measured factor acronyms: FFA, free fatty acid(s). The *BRR → GBR* is italicized since it was the original starting material to generate GBR but, it is technically not the GBR beverage control. ^2^ Means highlighted with an asterisk (*) are significantly different from the rice type control (GBR, BRR or WR) according to a Dunnett’s test at *p* < 0.05. Control and treatment means not connected by same letter are significantly different among them, according to a Tukey–Kramer HSD test at *p* < 0.05. z, y indicates the significant differences between the germinated GBR control versus the initial BRR used for germination; a, b, c indicates the differences among the GBR, BRR and WR controls; A,B,C indicates the differences among GBR, BRR and WR for the PWM treatments; X,Y,Z indicates the differences among GBR, BRR and WR for the PWM-SL treatments; r, s, t indicates the differences among GBR, BRR and WR for the PNZ treatments; and R,S,T indicates the differences among GBR, BRR and WR for the PNZ-SL treatment. The data represent the means from independent comparisons, where *n* = 3 or *n* = 6 ± standard deviation. ^3^ i.s. indicates insufficient sample to collect, whereas n.s. indicates not sampled.

**Table 3 foods-11-00220-t003:** Phytosterol esters (nonpolar lipids), free sterols, oryzanols and summed lipids/oils (mg/100 g) in the three beverages prepared from white, brown and germinated brown ‘Rondo’ rice, grouped by treatment, with commercial flour comparisons.

Treatments	StE (Phytosterol Esters)	Free Sterol	Oryzanol	Sum Classes (Recovered)
*BRR (→ GBR)* ^1^	*138.50 ± 14.12 z ^2^*	*35.30 ± 4.84 z*	*51.62 ± 11.84 z*	*3617.52 ± 756.10 z*
GBR (control)	97.29 ± 10.18 ya	26.92 ± 2.52 ya	25.10 ± 3.79 yb	2178.45 ± 152.70 yb
BRR (control)	102.05 ± 14.97 a	22.28 ± 1.75 b	35.67 ± 4.35 a	2673.07 ± 244.00 a
WR (control)	34.42 ± 1.81 b	3.35 ± 0.21 c	5.66 ± 0.96 c	751.17 ± 135.59 c
GBR, PWM	81.72 ± 5.02 *A	24.59 ± 0.42 A	18.11 ± 1.42 *B	1817.23 ± 77.78 *B
BRR, PWM	92.62 ± 9.91 A	22.66 ± 1.83 A	27.51 ± 1.39 A	2074.40 ± 116.70 *A
WR, PWM	17.25 ± 3.05 *B	2.40 ± 0.42 B	1.01 ± 0.25 *C	145.67 ± 21.34 *C
GBR, PNZ	29.02 ± 1.72 *t	8.74 ± 1.54 *t	5.84 ± 0.93 *t	554.98 ± 92.22 *t
BRR, PNZ	20.36 ± 5.89 *s	3.43 ± 0.53 *s	1.15 ± 0.06 *s	219.50 ± 36.39 *s
WR, PNZ	19.30 ± 0.82 *s	2.03 ± 0.27 s	0.36 ± 0.15 *s	142.55 ± 6.43 *s
GBR, PWM-SL	238.55 ± 20.89 *Z	50.84 ± 2.71 *Z	165.86 ± 11.43 *Z	4695.43 ± 448.29 *Z
BRR, PWM-SL	170.97 ± 16.21 *Y	43.62 ± 4.88 *Y	132.32 ± 12.56 *Y	3516.60 ± 52.56 *Y
WR, PWM-SL	i.s. ^3^	i.s.	i.s.	i.s.
GBR, PNZ-SL	n.s. ^3^	n.s.	n.s.	n.s.
BRR, PNZ-SL	137.21 ± 10.44 *T	41.91 ± 3.47 *T	69.20 ± 6.45 *T	3186.03 ± 254.20 *T
WR, PNZ-SL	33.42 ± 3.77 S	17.23 ± 2.53 *S	11.42 ± 1.07 *S	919.70 ± 33.94 S
CRF R7-150T	17.97 ± 2.98	7.03 ± 0.13	1.22 ± 0.06	278.31 ± 2.98
CRF RL-100	36.40 ± 16.45	8.57 ± 2.54	3.28 ± 0.43	862.96 ± 2.08

^1^ Treatment acronyms: BRR, brown ‘Rondo’ rice; GBR, germinated brown rice; PWM, post wet-milling; PNZ, post enzymes; PWM-SL, post wet-milling sieving loss; PNZ-SL, post enzymes sieving loss; WR; white rice and CRF, commercial rice flour. The *BRR (→ GBR)* is italicized since it was the original starting material to generate GBR but, it is technically not the GBR beverage control. ^2^ Means highlighted with an asterisk (*) are significantly different from the rice type control (GBR, BRR or WR) according to a Dunnett’s test at *p* < 0.05. Control and treatment means not connected by same letter are significantly different among them, according to a Tukey–Kramer HSD test at *p* < 0.05. z, y indicates the significant differences between the germinated GBR control versus the initial BRR used for germination; a, b, c indicates the differences among the GBR, BRR and WR controls; A,B,C indicates the differences among GBR, BRR and WR for the PWM treatments; X,Y,Z indicates the differences among GBR, BRR and WR for the PWM-SL treatments; r, s, t indicates the differences among GBR, BRR and WR for the PNZ treatments; R,S,T indicates the differences among GBR, BRR and WR for the PNZ-SL treatment. The data represent the means from independent comparisons, where *n* = 3 or *n* = 6 ± standard deviation. ^3^ i.s. indicates insufficient sample to collect, whereas n.s. indicates not sampled.

**Table 4 foods-11-00220-t004:** Percentage of compounds and compound classes recovered, based on the total lipids isolated in white, brown and germinated brown ‘Rondo’ rice beverages.

Treatments	% FFAs	% TAG	% TAGs and DAGs	% StE (Phytosterol Esters)	% Free Sterols	% Oryzanol
BRR (→ GBR) ^1^	2.43	87.45	91.34	3.83	0.98	1.43
GBR (control)	2.19	88.10	90.95	4.47	1.24	1.15
PWM	1.64	87.13	91.51	4.50	1.35	1.00
PNZ	9.97	79.07	82.17	5.23	1.58	1.05
PWM-SL	2.20	83.90	88.10	5.08	1.08	3.53
PNZ-SL	n.s. ^2^	n.s.	n.s.	n.s.	n.s.	n.s.
BRR (control)	1.99	88.15	92.03	3.82	0.83	1.33
PWM	0.55	88.93	92.57	4.47	1.09	1.33
PNZ	7.08	78.08	81.56	9.27	1.56	0.52
PWM-SL	0.96	85.29	89.17	4.86	1.24	3.76
PNZ-SL	6.75	80.94	85.45	4.31	1.32	2.17
WR (control)	69.63	22.35	24.59	4.58	0.45	0.75
PWM	12.94	64.74	72.88	11.84	1.65	0.70
PNZ	53.27	27.77	31.52	13.54	1.43	0.25
PWM-SL	i.s. ^2^	i.s.	i.s.	i.s.	i.s.	i.s.
PNZ-SL	41.83	44.59	51.43	3.63	1.87	1.24

^1^ Treatment acronyms: BRR, brown ‘Rondo’ rice; GBR, germinated brown rice; PWM, post wet-milling; PNZ, post enzymes; PWM-SL, post wet-milling sieving loss; PNZ-SL, post enzymes sieving loss and WR; white rice. Measured factor acronyms: FFA, free fatty acid(s); TAG, triacylglycerols; DAG, diacylglycerols and StE, phytosterol esters, including very nonpolar lipids. ^2^ n.s. indicates not sampled, whereas i.s. indicates insufficient sample to collect.

## Data Availability

The corresponding author will supply the calculated and statistical dataset (in spreadsheet format) upon reasonable and justified request.

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
