# Peer review of "Lipid Profiles in Preliminary Germinated Brown Rice Beverages Compared to Non-Germinated Brown and White Rice Beverages"

_foods, 2022, doi:10.3390/foods11020220_

Round 1

Reviewer 1 Report

The manuscript ‘Green processing, germinating and wet milling brown rice (Oryza sativa) for beverages: Lipid characterization’ is an interesting and deep study of using brown rice nad its health-promoting compounds in beverages. The study can be interesting for the potential reader, and the results can lead to interesting practical applications, but the paper needs to be improved.

COMMENTS:

The title of the paper should be reformulated and corresponds exactly to the analysis provided in the paper.

At the end of the introduction section clearly define the goal of the research provided in the paper. It should be clearly stated what the reader can expect reading the paper.

(l. 111-112) – there is a need to deliver short description of the ‘Sourcing ‘Rondo’ rice, methods used to de-hull, mill, sprout, and novel protocol to deliver a free-flowing soluble matrix through thermal softening, wet-milling, gelatinization and saccharification’. The reader has to be able to realize the main basis of the mentioned methodologies. The paper should be selfexplanatory, and the statement ‘were reported previously’ can’t be accepted

In ‘Materials and Methods’ section the information regarding number of samples, repetitions used for statistical analysis should be included

(l. 160) – ‘Ottawa sand’? Please explain

Table 1:

The table should be self-explanatory and restructured.  

Explain the treatment ‘BRR (->GBR)’.

The standard deviation should be added for all results corresponding to GBR, BRR and WR, as it is presented for commercial samples.

The way of the data presentation is strongly confused for the reader. The statistical significance is unreadable. The statistical significance should be unified across all samples analyzed – one set of letters.

Please explain if the statistical significance were analyzed across the samples (BRR, GBR, …) or across attributes (oleic, linoleic, …), or across row and columns together.

Table 2:

The table should be self-explanatory and restructured.  

Explain the treatment ‘BRR (->GBR)’.

The standard deviation should be added for all results corresponding to GBR, BRR and WR, as it is presented for commercial samples.

The way of the data presentation is strongly confused for the reader. The statistical significance is unreadable. The statistical significance should be unified across all samples analyzed – one set of letters.

Please explain if the statistical significance were analyzed across the samples (BRR, GBR, …) or across attributes (TAB, 1,3-DAG… ), or across row and columns together.

Table 3

The table should be self-explanatory and restructured.  

Explain the treatment ‘BRR (->GBR)’.

The standard deviation should be added for all results corresponding to GBR, BRR and WR, as it is presented for commercial samples.

Please consider replacing some tables by figures.

In ‘Conclusions’ section please underline the most importrant findings of your research.

Author Response

Foods (MDPI) foods-1403251 – Reviewer Comments – Corrections and Rebuttals

27 October 2021.

Foods MDPI

Dear Editors and Reviewers:

Thanks for the constructive criticism and for reviewing our manuscript.

Below, please find all Reviewer comments pasted into this document with line-by-line comments, edits described and/or rebuttals.

Per instructions, Track Changes has been used (for the most part).  Honestly, I might have forgotten to turn it on every time, but most salient changes are Tracked!

Please note that Track Changes was purposely NOT employed to fix the in-text citation format and the bibliography. Reference Manager did all those edits, and the brackets were manually added to the numerical format.

We look forward to any further comments or suggestions.

============*============*============*============

John C. Beaulieu, Robert A. Moreau, Michael J. Powell and Javier M. Obando-Ulloa. 2021. Green processing, germinating and wet milling brown rice (Oryza sativa) for beverages: Lipid characterization. Foods (MDPI). Submitted 15 September 2021. Conditionally Accepted 20 October 2021.

TITLE CHANGED

John C. Beaulieu, Robert A. Moreau, Michael J. Powell and Javier M. Obando-Ulloa. 2021. Lipid profiles in preliminary germinated brown rice beverages compared to non-germinated brown and white rice beverages

MDPI _ Reply review report_Reviewer #1_Round-1.pdf  (10-20-2021)

The manuscript ‘Green processing, germinating and wet milling brown rice (Oryza sativa) for beverages: Lipid characterization’ is an interesting and deep study of using brown rice nad its health-promoting compounds in beverages. The study can be interesting for the potential reader, and the results can lead to interesting practical applications, but the paper needs to be improved.

The title of the paper should be reformulated and corresponds exactly to the analysis provided in the paper.  Done.  New title is: “Lipid profiles in preliminary germinated brown rice beverages compared to non-germinated brown and white rice beverages”.

At the end of the introduction section clearly define the goal of the research provided in the paper. It should be clearly stated what the reader can expect reading the paper.  True.  Thanks for pointing this out.  It has been addressed.

(l. 111-112) – there is a need to deliver short description of the ‘Sourcing ‘Rondo’ rice, methods used to de-hull, mill, sprout, and novel protocol to deliver a free-flowing soluble matrix through thermal softening, wet-milling, gelatinization and saccharification’. The reader has to be able to realize the main basis of the mentioned methodologies. The paper should be self explanatory, and the statement ‘were reported previously’ can’t be accepted.  This has been added to the M&M, and since the other Reviewer also had similar concerns, we added the antimicrobial rinse information via the previously published M&M in (Beaulieu, Reed, Obando-Ulloa, & McClung, 2020b).

In ‘Materials and Methods’ section the information regarding number of samples, repetitions used for statistical analysis should be included. Clarification has been added (triplicate).

(l. 160) – ‘Ottawa sand’? Please explain.  We mix samples of plant material with Ottawa sand because it is recommended by the manufacturer of the Accelerated Solvent Extractor, Thermo Scientific Dionex.  Ottawa sand is a very clean inert uniformly shaped material that helps to prevent the sample from clumping and ensures good flow of solvents in the extraction vessel.  Research purpose, reagent grade Ottawa sand can be purchased from several companies (e.g. Resteck or Thermo Fisher Scientific).  As this is standard protocol, we do not believe a detailed explaination is needed in the manuscript but we added the manufacturer.

Table 1:

The table should be self-explanatory and restructured.

Explain the treatment ‘BRR (->GBR)’.  We added one parenthetical note in the beginning of the R&D (“3.1. Total Lipid and Proximate Analysis”) to clarify this designation:   …. BRR sprouted into GBR “(often designated as BRR → GBR, for clarity)” … and the GBR.

The standard deviation should be added for all results corresponding to GBR, BRR and WR, as it is presented for commercial samples.  We believe this would make the Tables very cumbersome and it is technically overkill considering that statistical significance is already illustrated for all presented data. Currently, we only placed the standard deviation for comparison samples (bottom of Tables 1 and 2) since it was not legitimate to compare commercial flour samples to our treatments through SAS.

The way of the data presentation is strongly confused for the reader. The statistical significance is unreadable. The statistical significance should be unified across all samples analyzed – one set of letters. Unfortunately, this is technically not possible since the WR cannot be germinated, we had one BR control, and we germinated another BR into GBR. As we used the same lot of rice, we analyzed data per rice beverage grouping based on it’s unique control, and then made comparisons across the whole experiment, per treatment, where there is not a control, per se. Subsequently, two sets of letters were purposely used. We believe the footnotes are rather detailed and differentiate that two comparisons via separate analyses were performed. Please read table footnotes better. Changing this format to only have one comparison in a table would imply additional tables would be need to be added.

            As clarification, some of the above text has been restated and added to the “2.3. Data Analysis and Statistics” of the M&M.

Please explain if the statistical significance were analyzed across the samples (BRR, GBR, …) or across attributes (oleic, linoleic,…), or across row and columns together. Again, please see the Table footnotes, and the comment immediately above.

      These two comments apply to all data presented, and all further reviewer comments on this subject.

Table 2:

The table should be self-explanatory and restructured.  I believe this comment might be in regard to the stats and letter designations being “cumbersome”?  In any event, since the M&M was updated for clarity, and the Table footnotes offer detailed explanations, we chose to leave the Tables, as is.

Explain the treatment ‘BRR (->GBR)’. Fixed, as explained above.

The standard deviation should be added for all results corresponding to GBR, BRR and WR, as it is presented for commercial samples.  Please see above comment to the same critique.

The way of the data presentation is strongly confused for the reader. The statistical significance is unreadable. The statistical significance should be unified across all samples analyzed – one set of letters. Please see above rebuttal and clarification remedy we accomplished, for the exact same critique.

Please explain if the statistical significance were analyzed across the samples (BRR, GBR, …) or across attributes (TAB, 1,3-DAG…), or across row and columns together. See above comment.

Table 3 – All comments and our rationale and/or rebuttals per identical questions herein have been stated above.

The table should be self-explanatory and restructured.

Explain the treatment ‘BRR (->GBR)’.

The standard deviation should be added for all results corresponding to GBR, BRR and WR, as it is presented for commercial samples. Please see above comment to the same critique.

Please consider replacing some tables by figures.  We prefer to keep the data as Tables.

In ‘Conclusions’ section please underline the most important findings of your research.  Since we do not see this being done in several recent Foods (MDPI) publications, this appears unnecessary?

MDPI _ Reply review report_Reviewer #2_Round-1.pdf  (10-20-2021)

References in the texts should be formatted according to authors guideline, and outdated references should be updated as well.   Done.

The introduction should be revised to be a little bit more concise. Agreed. Some unnecessary sentences were chopped out.

A flowchart with all the processing steps for the beverage should be added. Authors referred the reader to previously published M&M. This is published (Beaulieu, Reed, Obando-Ulloa, & McClung, 2020b), and also as a Supplemental figure in another related publication (Beaulieu, Reed, Obando-Ulloa, Boue, & Cole, 2020a) [citations 24 & 25]. The most salient points have been repeated in the paper.

Germination conditions are known to promote the growth of microbial species contaminating the grains.  This is true and we now refer back to an antimicrobial peracetic acid rinse that was employed, now cited in the M&M (Beaulieu, Reed, Obando-Ulloa, & McClung, 2020b). Furthermore, we rinsed the rice every 4 hours during imbibition and germination, so we too suspect that the levels of microbes are not really an issue. We published previously some microbial data (indicating increased yeast) in the above mentioned paper. However, we did not run microbial test herein. The other Reviewer also had this concern. Also, please see next comment.

Did the authors perform the count of molds, yeasts, and Enterobacteria after germination and evaluate how the following treatments affected them? This aspect is ???  Authors referred the reader to previous M&M whereby a peracetic acid anti-microbial rinse was accomplished. Microbial results from this aspect of the research were reported previously in (Beaulieu, Reed, Obando-Ulloa, & McClung, 2020b). Also, please see above comment.

L83: different, not differing.  Done.

L202: dwb should be explained the first time it is used.  Done.

Too many abbreviations in the text make the reading difficult. This is kind of true but we believe that the acronyms have been spelled out throughout the paper and again in all Tables.

L319: lower. Fixed.

Figures other than tables could be used to improve the quality of the manuscript.  We do not want to make changes and add figures. We prefer keeping the data presentation as Tables.

We hope these comments and clarifications suffice to appease the Reviewers.

We look forward to your replies.

Thanks for your consideration.

Sincerely,

Dr. John C. Beaulieu

USDA ARS

New Orleans, LA 70119

Beaulieu, J. C., Reed, S. S., Obando-Ulloa, J. M., Boue, S. M., & Cole, M. R. (2020a). Green processing, germinating and wet milling brown rice (Oryza sativa) for beverages: Physicochemical effects. Foods, 9(8), 1016.

Beaulieu, J. C., Reed, S. S., Obando-Ulloa, J. M., & McClung, A. M. (2020b). Green processing protocol for germinating and wet milling brown rice beverage formulations: Sprouting, milling and gelatinization effects. Food Science & Nutrition, 2020(8), 2445-2457.

Reviewer 2 Report

References in the texts should be formatted according to authors guideline, and outdated references should be updated as well.

The introduction should be revised to be a little bit more concise

A flowchart with all the processing steps for the beverage should be added.

Germination conditions are known to promote the growth of microbial species contaminating the grains. Did the authors perform the count of molds, yeasts, and Enterobacteria after germination and evaluate how the following treatments affected them? This aspect is

L83: different, not differing

L202: dwb should be explained the first time it is used

Too many abbreviations in the text make the reading difficult.

L319: lower

Figures other than tables could be used to improve the quality of the manuscript

Author Response

(The authors gave the same response as above.)
